# Implementation of a disability management policy in a large healthcare employer: a quasi-experimental, mixed-methods evaluation

Cameron A Mustard,[1,2] Kathryn Skivington,[3] Morgan Lay,[2] Marni Lifshen,[2] Jacob Etches,[4] Andrea Chambers[5]

[1]Dalla Lana School of Public Health, University of Toronto, Toronto, Ontario, Canada
[2]Research, Institute for Work & Health, Toronto, Ontario, Canada
[3]Public Health Sciences, University of Glasgow, Glasgow, UK
[4]Institute for Clinical Evaluative Sciences, Sunnybrook Health Science Centre, Toronto, Ontario, Canada
[5]Infection Prevention and Control, Public Health Ontario, Toronto, Ontario, Canada

**Correspondence to**
professor Cameron A Mustard;
cmustard@iwh.on.ca

## ABSTRACT

**Objective** This study describes the process and outcomes of the implementation of a strengthened disability management policy in a large Canadian healthcare employer. Key elements of the strengthened policy included an emphasis on early contact, the training of supervisors and the integration of union representatives in return-to-work (RTW) planning.

**Design** The study applied mixed methods, combining a process evaluation within the employer and a quasi-experimental outcome evaluation between employers for a 3-year period prior to and following policy implementation in January 2012.

**Participants** Staff in the implementation organisation (n=4000) and staff in a peer group of 29 large hospitals (n=1 19 000).

**Outcomes** Work disability episode incidence and duration.

**Results** Both qualitative and quantitative measures of the implementation process were predominantly positive. Over the 6-year observation period, there were 624 work disability episodes in the organisation and 8604 in the comparison group of 29 large hospitals. The annual per cent change in episode incidence in the organisation was −5.6 (95% CI −9.9 to −1.1) comparable to the annual per cent change in the comparison group: −6.2 (-7.2 to −5.3). Disability episode durations also declined in the organisation, from a mean of 19.4 days (16.5, 22.3) in the preintervention period to 10.9 days (8.7, 13.2) in the postintervention period. Reductions in disability durations were also observed in the comparison group: from a mean of 13.5 days (12.9, 14.1) in the 2009–2011 period to 10.5 days (9.9, 11.1) in the 2012–2014 period.

**Conclusion** The incidence of work disability episodes and the durations of work disability declined strongly in this hospital sector over the 6-year observation period. The implementation of the organisation's RTW policy was associated with larger reductions in disability durations than observed in the comparison group.

## INTRODUCTION

This study reports on the process and outcomes of the implementation of a strengthened disability management policy in a large Canadian acute care community hospital system employing more than 4300

### Strengths and limitations of this study

► A mixed-methods evaluation of the process and outcomes of the implementation of a disability management policy in a large employer.
► The evaluation includes a quasi-experimental design, comparing temporal changes in outcomes in the intervention organisation with a group of peer organisations.
► Stronger insights into the facilitators and barriers to organisational improvement would be available from the evaluation of implementation across multiple workplaces.

staff. In collaboration with union representatives, the organisation designed and implemented a return-to-work (RTW)/accommodation policy in 2012. The objective of the strengthened disability management policy was to reduce the incidence of work-related musculoskeletal disorders and improve organisational practices in the area of RTW and disability prevention, based on the principle that optimal workplace practices in the area of worker health will integrate efforts in the area of primary and secondary prevention.[1] The outcome objectives of the strengthened disability management policy were to reduce the incidence of work disability claims registered with the provincial workers' compensation authority by 25% and to reduce the total days of disability provided wage replacement benefits by 25% over the 3-year period 2012–2014. The evaluation described in this study had two process evaluation elements and one outcome evaluation component: (1) a qualitative case study examining the perspectives of supervisors, managers and RTW coordinators regarding the fidelity of the implementation of the RTW/accommodation policy, (2) repeated surveys of employees who had returned to

work following a disability episode and (3) a quasi-experimental design, comparing workers' compensation claim incidence and duration in the organisation to a comparison group of the 29 largest acute care hospitals in the province of Ontario.

In this section, we summarise the current state of evidence for the effectiveness of workplace practices to reduce the burden of disability arising from these disorders. Three recent publications (an editorial, a consensus statement and a literature review) have noted a gap in the research literature of studies evaluating the feasibility, effectiveness, costs and benefits of workplace RTW programmes.[2–4]

An early systematic review of research evidence for the effectiveness of workplace-based RTW interventions found strong evidence that work disability duration is significantly reduced by work accommodation offers and contact between workplaces and workers' healthcare providers and moderate evidence that work disability duration is reduced by interventions which include early contact with worker by workplace, ergonomic work site visits and presence of a RTW coordinator.[5] For these five intervention components, there was moderate evidence that they reduce costs associated with work disability duration.

The findings of this systematic review provided specific support for core programme features of workplace-based RTW programmes. These core programme features have been distilled to seven principles of return work[6]: (1) the workplace has a strong commitment to health and safety which is demonstrated by the behaviours of the workplace parties, (2) the employer makes an offer of modified work (also known as work accommodation) to injured/ill workers so they can return early and safely to work activities suitable to their abilities, (3) RTW planners ensure that the plan supports the returning worker without disadvantaging coworkers and supervisors, (4) supervisors are trained in work disability prevention and included in RTW planning, (5) the employer makes an early and considerate contact with injured/ill workers, (6) someone has the responsibility to coordinate RTW and (7) employers and healthcare providers communicate with each other about the workplace demands as needed, and with the worker's consent.

Franche *et al* recently reported on the factors associated with an offer of work accommodation in a cohort of more than 500 Ontario workers disabled by a musculoskeletal injury.[7] Approximately 60% of workers received an offer of work accommodation in this cohort and workers were more likely to receive a work accommodation offer from their employer, if the workplace was unionised or if the worker perceived strong disability prevention policies and practices in their workplace. In this large Ontario cohort, workplace factors were an important determinant of an offer of work accommodation. Workers who reported their workplace to be in the top third of a standardised measure of organisational policies and practices related to RTW[8 9] were three times more likely to

receive an offer of work accommodation than workers who reported their workplace to be in the bottom third. Further work on this cohort has confirmed that organisational policies and practices in RTW are central factors in workplace disability prevention performance.[10] Workers disabled by a musculoskeletal disorder who reported strong workplace organisational policies and practices in RTW were more likely to have returned to work at 6 months (OR=1.85; 95% CI 1.12 to 3.07) and at 12 months (OR=2.30; CI 1.31 to 4.04).

There are challenges in consistently administering a workplace disability management programme.[11–13] In many workplaces, an important proportion of disabled employees experience a variable and often undesirable RTW course including extended or intermittent work disability that results in significant individual, employer and societal costs.[13–15] Some of this inconsistency can be attributed to the sometimes competing goals of employers (eg, minimise costs) and workers (eg, recovery from illness), a lack of coordination and communication among work disability stakeholders (eg, workers, healthcare providers, employers and workers compensation insurers) and gaps in the provision of adequate accommodations.[14] In addition, supervisors and coworkers often have important responsibilities for managing different phases of the RTW process and their interaction with injured workers played an important role in determining work disability outcomes.[16–18]

Of relevance to the objectives of this study, there is an important research literature on the factors that support successful implementation of disability management practices across organisations. An initiative in Denmark to systematically implement a national RTW programme standard in 21 Danish municipalities identified substantial variation in disability outcomes following implementation.[19 20] Evidence documented in a comprehensive process evaluation identified corresponding variation in the fidelity with which individual municipalities implemented five key dimensions of the RTW programme and that weak implementation was associated with poor disability management outcomes.[21] The current state of the research evidence concerning the effectiveness of organisational policies and practices to protect worker health points to the need for high-quality studies of the effectiveness of multicomponent disability prevention programmes.

## THE SETTING
The employer is one of Ontario's largest multisite acute care community hospital systems with a US$390 million annual operating budget and a staff complement of approximately 4300. Employees are represented by three unions.

The organisation recognised limitations in the integrity of occupational health and safety (OHS) policies and practices across the care sites in the system, including an underdeveloped disability management policy. There had

been conflicts between management and representatives of workers over RTW practices, and the labour relations environment was frictional.

## THE RTW/ACCOMMODATION POLICY

Over a 12-month period, the RTW/accommodation policy was developed jointly by the organisation's management and union representatives, with contributions from not-for-profit external advisors. The evaluation research team did not participate in the development of RTW/accommodation policy. The policy was closely anchored to guidance provided by research evidence,[5 22 23] including an emphasis on early contact, the designation of disability case managers, the integration of supervisors in the development of RTW plans and the provision of education and training to managers and supervisors. A distinctive component of the policy was a specific role defined for union representatives as 'return-to-work coordinators' (RTWC).[24] RTWCs participated in RTW planning meetings, along with the employee, the organisation's disability case managers and the employee's supervisor. Details on the specific components of the RTW/accommodation policy have been published elsewhere.[24] Implementation of the renewed RTW/accommodation policy commenced in January 2012.

The outcome objectives of the strengthened disability management policy were to reduce the incidence of work disability claims registered with the provincial workers' compensation authority by 25% and to reduce the total days of disability provided wage replacement benefits by 25% over the 3-year period 2012–2014.

## METHODS
### Study design
This study applied a mixed methods protocol consisting of two components. The process evaluation component had two elements: (1) a qualitative study of the perspectives of managers and supervisors concerning the fidelity of the implementation of the return to work/accommodation policy and (2) the administration of two annual surveys of a sample of the National Health Service (NHS) staff who had returned to work following a disability episode. The outcome evaluation component applied a quasi-experimental design, comparing workers' compensation claim incidence and duration in the organisation to a peer group of 29 large Ontario hospitals for a 3-year period prior to and following January 2012.

### Process evaluation: supervisors and RTW coordinators
All managers and RTWCs who had been involved in a RTW episode in 2012 (n=104) were invited to participate in in-person semi-structured interviews. Interviews were completed with 21 managers and nine union RTWCs. Interviews followed a topic guide with general themes about the RTW policy: roles and responsibilities under the process; early contact practices; how the RTW plan was developed; how RTW is monitored; access to confidential

information and general overall assessment of the positive and negative elements of the programme. A thematic analysis of interview transcripts was performed.

### Process evaluation: employees experiencing a disability episode
All employees who participated in a RTW plan following a disability episode in 2013 and 2014 were invited to complete an internet-administered 14 item questionnaire. In both years, the response rate was approximately 30%. The 14-item questionnaire asked respondents for their perception of the accommodation process using a five-point Likert scale (strongly agree to strongly disagree). Respondents were also invited to contribute free text responses on their overall experience, initial contact with the NHS's OHS staff, the RTW transition and issues around health information confidentiality. Chi-square tests were applied to test for differences between 2013 and 2014 in the proportion of respondents who endorsed each of the 14 questionnaire items.

## OUTCOME EVALUATION
The outcome evaluation component applied a quasi-experimental design, comparing workers' compensation claim incidence and duration in the organisation to a peer group of the 29 largest Ontario hospitals over the 6-year period 2009–2014.

Electronic abstracts of administrative records of lost-time and no lost-time compensation claims were obtained from the Workplace Safety and Insurance Board (WSIB), the publicly administered, single-payer work disability insurer in Ontario. The outcome evaluation component of this study is based on two measures obtained from administrative data: (1) the incidence rate of lost-time and no lost-time claims and (2) the mean duration of lost-time disability episodes arising from work-related disorders. The incidence rates for lost-time and no lost-time claims were computed on an annual basis with denominator information of person-years of employment obtained from WSIB insurance records of the organisation and the peer group of 29 large hospitals. The duration of disability was computed for each lost-time claim based on days of wage replacement benefits and the mean duration of disability, in days, computed for all claims on an annual basis. Confidence intervals (95%) for incidence rates and disability day rates were estimated under an assumption of a Poisson distribution. We estimated average annual per cent change in incidence rates using negative binomial regression with the log of person-years of employment as an offset. We estimated annual rates of RTW at intervals of 7, 14, 30 and 60 days using piecewise exponential survival models in SAS PROC GENMOD and tested for differences in trends of RTW durations between the organisation and the peer group.[25] The outcome evaluation component of this study was designed to have sufficient power to detect a reduction in the incidence of workers' compensation claims from 75

per 1000 full-time equivalents (2009–2011) to 65 per 1000 full-time equivalents over the period 2012–2014.

## ETHICS

Ethical approval was obtained from the Health Sciences Research Ethics Board, University of Toronto (Protocol 28503). Written informed consent was obtained from all survey and interview participants.

## RESULTS

The implementation of the RTW/accommodation policy was the responsibility of the organisation's OHS department. The progress of the implementation was monitored by a labour-management steering committee, chaired by the organisation's vice-president, human resources. Steering committee membership include union local leadership, OHS department staff and external members from a number of provincial labour unions. The steering committee met four to six times per year during the 3-year implementation period documented in this study. At the invitation of the labour and employer members of the organisation, the research team was also a member of the steering committee and periodically provided reports on aspects of the process and outcomes of the policy implementation.

### Process evaluation component

Both qualitative and quantitative measures of the implementation process were predominantly positive. Supervisors, managers and RTWCs who participated in semi-structured interviews in 2013 noted a number of achievements.[24] Informants offered the view that OHS practice had improved since the implementation of the policy and spoke of the clear internal communication around the RTW/accommodation policy to all levels of staff in the organisation. Informants also perceived a renewed emphasis on returning disabled workers to their preinjury positions as quickly as possible and that the staff in the OHS department were well respected. There was recognition that the quality and consistency of disability management practices were improving, that the collaboration between the employer and the labour unions was strengthened and that the RTW planning meeting was an important opportunity for establishing clarity for the goals of individual RTW plans.

Managers, supervisors and RTW coordinators also identified a number of opportunities for improvement. Informants encouraged a greater emphasis on timely access to information about the status of individual disability episodes, and identified a number of opportunities to improve the consistency of case management across disability episodes. Informants also described a number of areas where the roles and responsibilities RTW coordinators and supervisors were ambiguous and emphasised the importance of providing orientation to newly appointed personnel. Interview participants also

acknowledged that the OHS department had a high case load and constrained resources.

### Employees experiencing a disability episode

Approximately 30% of employees who participated in a return to work plan accepted the invitation to complete an internet-administered 14-item questionnaire. The majority of respondents who returned to work in 2013 (n=78) and in 2014 (n=54) reported positive assessments of the RTW process (table 1). Employees gave strong endorsement to the purpose and outcome of the RTW planning meeting and expressed satisfaction with the protection of personal health information. Respondents in 2014 reported stronger endorsement of a number of programme dimensions compared with respondents in 2013. For example, a higher proportion of respondents in 2014 were satisfied with modified duty arrangements and a smaller proportion of respondents in 2014 reported delays in arranging the initial RTW planning meeting. There were no significant differences in perceptions of the RTW process between respondents whose disability was attributed to a work-related cause, and respondents whose disability was attributed to a non-occupational cause.

Free text responses provided by survey respondents were approximately balanced between positive and negative statements. Themes that were emphasised in the free text responses included the importance of supportive, compassionate and respectful interactions with OHS staff, the importance of support from managers and supervisors and the value of efficient and timely communication with OHS staff. Respondents also emphasised the value of monitoring the progress of accommodations over the course of the disability episode and the importance of aligning modified duties to clinical recommendations.

### Outcome evaluation component

Over the 6-year observation period, there were 1916 total compensation claims in the organisation, and 29 974 compensation claims in the comparison group of 29 large hospitals (table 2). The incidence rates for lost-time claims and for no lost-time claims were substantially higher in the organisation relative to the comparison group. In both the organisation and the comparison group, the decline in the incidence of lost-time claims between the 2009–2011 period and the 2012–2014 period was statistically significant. The annual per cent change in the lost-time claim incidence rate in the organisation was −5.6 (95% CI −9.9 to −1.1) comparable to the annual per cent change in the comparison group: −6.2 (95% CI -7.2 to −5.3).

Disability durations also declined in the organisation, from a mean of 19.4 days (16.5, 22.3) in the preintervention period to 10.9 days (8.7, 13.2) in the postintervention period (table 3). Reductions in disability durations were also observed in the comparison group: from a mean of 13.5 days (12.9, 14.1) in the 2009–2011 period to 10.5 days (9.9, 11.1) in the 2012–2014 period.

**Table 1** RTW employees' perceptions of the disability management process five-point Likert scale (strongly disagree to strongly agree).

| | Per cent of employees agreeing with statement | | |
|---|---|---|---|
| | **2013** | **2014** | |
| | **(n=78)** | **(n=54)** | |
| **Supported through the process:** | | | $\chi^2$ |
| My supervisor/manager supported my transition back to work | 70% | 76% | 0.49 |
| My coworkers supported by transition back to work | 76% | 78% | 0.78 |
| My absence and my RTW experience was in keeping with my employer's core values of compassion professionalism and respect | 53% | 67% | 0.11 |
| **Satisfaction with the process:** | | | |
| Overall, I was satisfied with my union's support in my RTW | 72% | 69% | 0.69 |
| Overall, I was satisfied with the employer's support in my RTW | 58% | 68% | 0.21 |
| I was satisfied with arrangement to modify my job duties and/or work hours during the transition back to full regular duties | 60% | 81% | 0.01 |
| **Confidence about the process:** | | | |
| *Timeliness* | | | |
| There were delays in arranging my RTW planning meeting | 42% | 28% | 0.09 |
| Documentation from my healthcare provider supported the timeliness of my RTW | 83% | 91% | 0.22 |
| I was contacted shortly after my absence began by the occupational health and safety department | 76% | 80% | 0.60 |
| *The RTW meeting* | | | |
| I understood the purpose of the RTW planning meeting | 87% | 87% | 0.98 |
| The RTW planning meeting identified options to modify or redesign my regular work and/or hours | 67% | 83% | 0.03 |
| I was an active participant in the RTW planning meeting | 79% | 82% | 0.77 |
| Planning my RTW was a team effort | 64% | 80% | 0.05 |
| *Confidentiality* | | | |
| Confidential information about my health was protected | 72% | 85% | 0.07 |

RTW, return-to-work.

In the preintervention period, the proportion of disability episodes in the organisation that had returned to work at 7, 30 and 60 days was substantially lower than observed outcomes in the peer group. As depicted in figure 1, the proportion of disability episodes that had returned to work at 7 days improved in both the

**Table 2** Incidence of compensation claims, organisation compared with 29 peer hospitals, 2009–2014

| | 2009–2011 | | | | 2012–2014 | | | |
|---|---|---|---|---|---|---|---|---|
| | **N** | **FTE** | **Rate/1000** | **95% CI*** | **N** | **FTE** | **Rate/1000** | **95% CI*** |
| **Lost-time claims** | | | | | | | | |
| Organisation | 362 | 12 062 | 30.0 | 27.1 to 33.3 | 287 | 12 849 | 22.3 | 19.9 to 25.1 |
| Peer hospitals | 4983 | 357 104 | 14.0 | 13.6 to 14.3 | 3986 | 379 150 | 10.5 | 10.2 to 10.8 |
| **No lost-time claims** | | | | | | | | |
| Organisation | 631 | 12 062 | 52.3 | 48.4 to 56.6 | 636 | 12 849 | 49.5 | 45.8 to 53.5 |
| Peer hospitals | 9835 | 357 104 | 27.5 | 27.0 to 28.1 | 11 170 | 379 150 | 29.5 | 28.9 to 30.0 |
| **Total claims** | | | | | | | | |
| Organisation | 993 | 12 062 | 82.3 | 77.4 to 87.6 | 923 | 12 849 | 71.8 | 67.3 to 76.6 |
| Peer hospitals | 14 818 | 357 104 | 41.5 | 40.8 to 42.2 | 15 156 | 379 150 | 40.0 | 39.3 to 40.6 |

*The CI assumes a Poisson distribution in the numerator.

**Table 3** Lost-time compensation claim benefit duration, organisation compared with 29 peer hospitals, 2009–2014 average benefit days per claim, first 90 days postinjury

| | 2009–2011 | | | 2012–2014 | | |
|---|---|---|---|---|---|---|
| | Number of claims | Average benefit days | 95% CI | Number of claims | Average benefit days | 95% CI |
| **Lost-time claims** | | | | | | |
| Organisation | 362 | 19.4 | 16.5 to 22.3 | 287 | 10.9 | 8.7 to 13.2 |
| Peer hospitals | 4983 | 13.5 | 12.9 to 14.1 | 3986 | 10.5 | 9.9 to 11.1 |

organisation and the comparison group over the 6-year observation period, with a stronger improvement in the organisation. A test for difference in the trend of improvement between the organisation and the comparison group approached statistical significance (p=0.07).

## DISCUSSION

This study has described the process and outcomes of a 3-year effort to implement a strengthened disability management policy in a large healthcare employer. Both the process and outcome findings of this study document a successful implementation of the strengthened policy. Managers, supervisors and RTW coordinators broadly endorsed the strengthened policy. Employees who returned to work following a disability episode provided predominantly positive assessments across all dimensions of the disability management process. The annual per cent reduction in lost-time claims in the organisation was similar to the reductions observed in a peer group of 29 hospitals over a 6-year period, consistent with a broad trend of a decline in the incidence of work-related injury and illness in all economic sectors in this jurisdiction.[26] Independent of this trend in declining incidence, the implementation of the organisation's strengthened

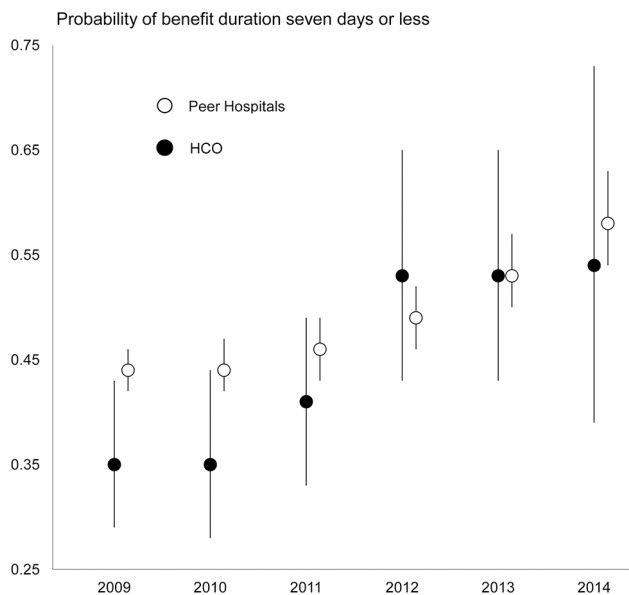

**Figure 1** Trend in disability benefit duration, 7 days or less organisation compared with 29 peer hospitals, Ontario 2009–2014 work-related lost-time compensation claims.

RTW policy was associated with greater improvements in disability durations than observed in the peer group.

Despite the broad consensus that workplace polices and practices are a central determinant of RTW outcomes, the literature on the implementation of disability management programmes in individual workplaces is limited and represents a research gap.[2 3] This study contributes to addressing this research gap. While the observational study was not suitable for determining which elements of the strengthened disability management policy were most responsible for the improvements in disability durations, the favourable process and outcome measures reported in this study indicate that it is feasible to successfully implement a moderately complex multicomponent disability management policy. From the perspective of employees who returned to work following a disability episode, it appears that all elements of the RTW process were meaningful, including early contact, the process and outcome of the RTW planning meeting and the support of supervisors and union representatives in the transition back to work. The collaborative model adopted by this organisation, where representatives of a worker's union participate in RTW planning, appears to have functioned effectively. RTW coordinators and disability management staff developed respect for the consistency and transparency of the collaborative process. And we note there was continuity and consistency in the organisational leadership supporting this initiative.

The evaluation design combining qualitative and quantitative methods to document the process and outcome of the policy implementation was a strength of this study. The inclusion of a comparison group of peer hospitals operating in the same geographic jurisdiction provided an accurate calibration of the magnitude of change achieved by the organisation.

We would briefly note the following limitations. Although the evaluation design did include a quasi-experimental component, the study has reported on the process and outcome of change in a single organisation. Stronger insights into the facilitators and barriers of organisational improvement in disability management practices would be available from evaluations of implementation process and outcome across multiple workplaces, where the attributes of organisations that successfully accomplish the programme objective can be compared with organisations that are less successful.[21 27 28] While this study has compared the incidence and durations of disability

episodes arising from work-related conditions between the organisation and a large hospital peer group, we were unable to acquire similar administrative records of work disability incidence and duration for disability episodes not attributed to work exposures.

The organisational change initiative reported in this study was designed jointly by labour and management representatives within a large Ontario public sector employer, with technical assistance provided by not-for-profit external agencies with expertise in workplace quality improvement initiatives. This organisation change initiative was accomplished with the technical and human resources available to the employer and did not require significant external resources. Relative to the outcome objectives established at the initiation of this quality improvement initiative, the organisation accomplished a reduction of approximately 15% in the incidence of work disability claims and achieved an approximate 50% reduction in average disability episode durations over the 3-year period following the implementation of the strengthen disability management policy. Replication of the collaborative model describes in this study may be considered in other healthcare settings in this jurisdiction.

**Contributors** CAM conceptualised and designed the study, interpreted the data and wrote the first draft of the paper. KS, ML, AC and ML contributed to data collection and data interpretation. JE contributed to the analysis of the study data and provided interpretation. All authors contributed to the review and revision of the paper.

**Competing interests** None declared.

**Ethics approval** University of Toronto.

**Provenance and peer review** Not commissioned; externally peer reviewed.

**Data sharing statement** No additional data available.

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
