## [Reviewer comments · BMJ Open]

ARTICLE DETAILS

TITLE (PROVISIONAL)	Implementation of a disability management policy in a large healthcare employer: a quasi-experimental, mixed-methods evaluation
AUTHORS	Mustard, Cameron; Skivington, Kathryn; Lay, Morgan; Lifshen, Marni; Etches, Jacob; Chambers, Andrea

VERSION 1 - REVIEW

REVIEWER	William S. Shaw Liberty Mutual Research Institute for Safety, USA I am one of 33 adjunct scientists who serve in an advisory role at the Institute for Work & Health (the first author's institution). However, I receive no direct compensation or research funding from the Institute, and I am not working directly with any of the authors. Thus, I feel I can provide sound and objective professional judgment about the submitted manuscript without any competing interest.
REVIEW RETURNED	02-Nov-2016

GENERAL COMMENTS	Manuscript Review, BMJ Open 10/31/2016 "Implementation of a disability management policy in a large healthcare employer: a quasi-experimental evaluation" This manuscript evaluates the success of implementing an improved, evidence-based disability management (DM) policy in a large (4,000-employee) hospital system over a 6-year period (3 years pre, 3 years post). Comparison data from a larger group of 29 large hospitals (N = 119,000 employees) were used to gauge benefits of the new policies in the context of industry-wide trends elsewhere. The new DM policies consisted of promoting early contact with absent employees, designating union representatives as disability case managers, and better training and integration of supervisors in return-to-work (RTW) planning. The primary findings were a 6 percent reduction in injury rates (a trend that was also apparent in the comparison hospitals) and a 44 percent reduction in disability duration (compared with a 22 percent reduction in the comparison hospitals). A parallel process evaluation also supported the acceptability and uptake of new practices within the organization. The authors conclude that the implementation of new DM policies are associated with reductions in disability duration that outpace industry trends. Underperforming RTW policies and unnecessary disability can represent a serious quality-of-life issue for employees and a significant cost and organizational issue for employers. Published reports involving well-designed researcher-employer collaborations to implement and evaluate new DM policies are extremely sparse.
---

The authors are to be congratulated for having overcome the significant organizational hurdles that are often encountered in this type of research involving industry partnerships. Though quasi-experimental designs have inherent limitations, this study provides an excellent opportunity to quantify effects of adopting new organizational policies and to simultaneously describe observations related to implementation and feasibility.

The study itself has many strengths: an organizational approach to intervention, reliance on evidence-based DM practices from the scientific literature, access to complete WC administrative data for the organization and for comparable organizations, integration of union representatives in the DM program, inclusion of quantitative and qualitative metrics for process evaluation, and a straightforward analytic approach consistent with study aims. However, there are a number of changes that could be made to improve the quality and impact of the manuscript. Detailed comments follow:

1. Abstract: The term “strengthened disability management policy” is the only language in the abstract describing the organizational policy change that was the focus of the study, and this terminology might be misinterpreted as improvements in administrative tracking systems or written policies of the organization. If word length allows, the abstract should clarify that changes involved supervisor training and integration in RTW planning, installing union representatives as RTW coordinators, and encouraging earlier contact with absentee workers. If this is too many words, another approach would be to replace “strengthened” with “evidence-based” or “proactive”. This will be a critical issue for this manuscript to be found and included in future systematic reviews.

2. The setting (page 8, first paragraph): It would be helpful here for readers to understand the nature of the collaboration between the researcher group and the subject hospital. How was this collaboration first established, and how was this relationship managed over the course of the study, especially with regard to inclusion of Union representatives and other stakeholders? Was the study part of a regulatory mandate imposed on the organization? It’s unclear whether researchers had any input into the actual policy changes that were instituted, and this participation should be clarified.

3. Methods (page 8, paragraph 2): Only one paragraph describes the organizational intervention components, and this does not provide sufficient detail of the scope and magnitude of the organizational changes. More specifics with regard to training, assigned responsibilities, altered processes, and methods of communication are needed so that these beneficial organizational changes could be reasonably replicated by another employer.

4. Results (page 14, last paragraph): Computation of disability days per 100 full-time equivalents before and after policy changes (as an alternative to average disability duration computations) might provide another illustrative metric that would be less confounded by injury reporting rates and severe outliers.

5. Methods (page 8, paragraph 2): Although the data analyses presume that all policies were adopted at a single point in the project timeline, it seems that actual implementation would have taken considerably longer in this large organization. Can more be said about the roll-out process and the approximate time frame for various policy elements? Impact of policy changes may be underestimated if implementation is more of a slow developing process and cultural change within the organization.

	6. Results (page 14, paragraph 2): The manuscript states that free text responses from survey respondents were evenly divided between positive and negative statements, but only examples of positive comments are listed. To be fair, examples of negative comments should be provided. 7. Discussion (page 16, paragraph 2): One perhaps unexpected but very important finding from the study is the sizable reduction in injury rates and disability costs in the comparison group of hospitals. Given the large size of this group, this appears to be an overall industry trend and is not likely to be a chance finding. Can more explanation be provided in the Discussion about this positive industry trend? Can it be explained in terms of more widespread changes to policies or regulations affecting the health care sector in Canada? 8. Discussion (page 17, paragraph 2): It's notable that the volunteer organization showed less-than-average performance on disability costs prior to the intervention and that the post-intervention results merely matched the average performance of the comparison peer group. This begs the question as to whether the policy changes merely matched current industry standards (as opposed to testing an innovative strategy). Can anything more detailed be said about the ways in which the organization's policies may have been subpar initially (without alienating the host organization)? This information might be insightful for other hospitals contemplating similar policy changes. 9. Methods (page 8, last paragraph): Given the focus of policy changes on DM practices, the hypothesis of reduced injury rates seems unfounded. Please explain or clarify.
--	--

REVIEWER	Marc White University of British Columbia Canada
REVIEW RETURNED	07-Apr-2017

GENERAL COMMENTS	The paper is very well written. It avoids specialized language and is easy to understand across different audiences. The conceptual framework and research design demonstrates the benefits of using mixed methods. The paper lacks information about process evaluation and the potential influence of the research design and feedback process on the steering committee and their commitment /engagement in the process. (Having attended a conference with a presentation of this work I noted the "participatory action research" component that may be a contributing factor to the evaluation success and important part of the implementation process. The feedback process to the steering committee (and leadership) is missing from methods, discussion, outcomes/limitations and this limits the ability for replication or the appreciation of the role of the evaluating process as a potent contributor to the implementation process.
--

VERSION 1 – AUTHOR RESPONSE

Reviewer: 1

Reviewer Name: William S. Shaw

Institution and Country: Liberty Mutual Research Institute for Safety, USA Competing Interests: I am one of 33 adjunct scientists who serve in an advisory role at the Institute for Work & Health (the first author's institution). However, I receive no direct compensation or research funding from the Institute,

and I am not working directly with any of the authors. Thus, I feel I can provide sound and objective professional judgment about the submitted manuscript without any competing interest.

This manuscript evaluates the success of implementing an improved, evidence-based disability management (DM) policy in a large (4,000-employee) hospital system over a 6-year period (3 years pre, 3 years post). Comparison data from a larger group of 29 large hospitals (N = 119,000 employees) were used to gauge benefits of the new policies in the context of industry-wide trends elsewhere. The new DM policies consisted of promoting early contact with absent employees, designating union representatives as disability case managers, and better training and integration of supervisors in return-to-work (RTW) planning. The primary findings were a 6 percent reduction in injury rates (a trend that was also apparent in the comparison hospitals) and a 44 percent reduction in disability duration (compared with a 22 percent reduction in the comparison hospitals). A parallel process evaluation also supported the acceptability and uptake of new practices within the organization. The authors conclude that the implementation of new DM policies are associated with reductions in disability duration that outpace industry trends.

Underperforming RTW policies and unnecessary disability can represent a serious quality-of-life issue for employees and a significant cost and organizational issue for employers. Published reports involving well-designed researcher-employer collaborations to implement and evaluate new DM policies are extremely sparse. The authors are to be congratulated for having overcome the significant organizational hurdles that are often encountered in this type of research involving industry partnerships. Though quasi-experimental designs have inherent limitations, this study provides an excellent opportunity to quantify effects of adopting new organizational policies and to simultaneously describe observations related to implementation and feasibility.

The study itself has many strengths: an organizational approach to intervention, reliance on evidence-based DM practices from the scientific literature, access to complete WC administrative data for the organization and for comparable organizations, integration of union representatives in the DM program, inclusion of quantitative and qualitative metrics for process evaluation, and a straightforward analytic approach consistent with study aims. However, there are a number of changes that could be made to improve the quality and impact of the manuscript. Detailed comments follow:

1. Abstract: The term “strengthened disability management policy” is the only language in the abstract describing the organizational policy change that was the focus of the study, and this terminology might be misinterpreted as improvements in administrative tracking systems or written policies of the organization. If word length allows, the abstract should clarify that changes involved supervisor training and integration in RTW planning, installing union representatives as RTW coordinators, and encouraging earlier contact with absentee workers. If this is too many words, another approach would be to replace “strengthened” with “evidence-based” or “proactive”. This will be a critical issue for this manuscript to be found and included in future systematic reviews.

We thank the reviewer for this advice and have revised the Objective statement in the abstract to include the following:

Key elements of the strengthened policy included an emphasis on early contact, the training of supervisors and the integration of union representatives in return-to-work planning.

2. The setting (page 8, first paragraph): It would be helpful here for readers to understand the nature of the collaboration between the researcher group and the subject hospital. How was this collaboration first established, and how was this relationship managed over the course of the study, especially with regard to inclusion of Union representatives and other stakeholders? Was the study

part of a regulatory mandate imposed on the organization? It's unclear whether researchers had any input into the actual policy changes that were instituted, and this participation should be clarified.

In response to this recommendation, we have incorporated brief revisions at two locations in the manuscript

p7/para2:

The evaluation research team did not participate in the development of return-to-work / accommodation policy.

p11/para1:

At the invitation of the labour and employer members of the organization, the research team was also a member of the steering committee and periodically provided reports on aspects of the process and outcomes of the policy implementation.

3. Methods (page 8, paragraph 2): Only one paragraph describes the organizational intervention components, and this does not provide sufficient detail of the scope and magnitude of the organizational changes. More specifics with regard to training, assigned responsibilities, altered processes, and methods of communication are needed so that these beneficial organizational changes could be reasonably replicated by another employer.

We appreciate the suggestion of the reviewer. We have previously published a paper providing details on the specific components of the revised policy. We have referenced this paper in the manuscript as follows:

Details on the specific components of the return-to-work / accommodation policy have been published elsewhere (24).

4. Results (page 14, last paragraph): Computation of disability days per 100 full-time equivalents before and after policy changes (as an alternative to average disability duration computations) might provide another illustrative metric that would be less confounded by injury reporting rates and severe outliers.

We agree in principle with the reviewer's recommendation. Our decision to report the average duration of disability episodes was guided by the interest in most clearly documenting the influence of the return-to-work / accommodation policy. As noted in Table 2, the incidence of episodes of disability attributed to work-related causes was almost twice as high in the organization compared to the comparison group of 29 large hospitals. A measure of disability days per 100 full-time equivalents would combine these differences in incidence with differences in average duration. We think the decision to report the average disability episode duration provides the clearest information on the program outcome and we have not incorporated a revision.

5. Methods (page 8, paragraph 2): Although the data analyses presume that all policies were adopted at a single point in the project timeline, it seems that actual implementation would have taken considerably longer in this large organization. Can more be said about the roll-out process and the approximate time frame for various policy elements? Impact of policy changes may be underestimated if implementation is more of a slow developing process and cultural change within the organization.

This is an excellent point. And a challenge to provide a precise response. Our interest in both process measurement and outcome measures was to acquire information that could document changes over time. So, for example, the process measures reported in Table 1 do document (we think) a trend

towards improvement in employee's perception of the disability management process between a measure collected approximately 12 months post-implementation and a measure collected 24 months post-implementation. In contrast, the outcome measure reported in Figure 1 would suggest that the reductions in disability episode duration occurred relatively abruptly in the initial phase of implementation. We absolutely see the merit of the reviewer's interest. Unfortunately, beyond the measures reported in the paper (as described in this paragraph), we cannot confidently respond to this interest to document or describe the fidelity of program implementation over time.

6. Results (page 14, paragraph 2): The manuscript states that free text responses from survey respondents were evenly divided between positive and negative statements, but only examples of positive comments are listed. To be fair, examples of negative comments should be provided.

We understand the reviewer's interest on this point. Our intent, in the language used in para2/p14 was to neutrally describe the themes arising from free text responses provided by survey respondents (who had returned-to-work following a disability episode). So for example, the phrase 'free text responses... emphasized the importance of support from managers and supervisors' was written to acknowledge that both negative and the positive statements had been provided by respondents. As we state in the opening sentence of para2/p14, these responses were approximately balanced between positive and negatives statements. We trust this clarification is helpful. We have not revised the paragraph language.

7. Discussion (page 16, paragraph 2): One perhaps unexpected but very important finding from the study is the sizable reduction in injury rates and disability costs in the comparison group of hospitals. Given the large size of this group, this appears to be an overall industry trend and is not likely to be a chance finding. Can more explanation be provided in the Discussion about this positive industry trend? Can it be explained in terms of more widespread changes to policies or regulations affecting the health care sector in Canada?

We thank the reviewer for this point. We have responded with a slight revision to para1/p16 to note the broad secular trend in this sector and in all sectors to a reduction in the frequency of work-related injury and illness, and have added a citation documenting this trend in this jurisdiction.

8. Discussion (page 17, paragraph 2): It's notable that the volunteer organization showed less-than-average performance on disability costs prior to the intervention and that the post-intervention results merely matched the average performance of the comparison peer group. This begs the question as to whether the policy changes merely matched current industry standards (as opposed to testing an innovative strategy). Can anything more detailed be said about the ways in which the organization's policies may have been subpar initially (without alienating the host organization)? This information might be insightful for other hospitals contemplating similar policy changes.

We think the reviewer's interpretation is accurate: the organization's disability management policy (and practices) were weaker relative to the peer group norms prior to the policy change, and in fact were the motivation for the change. We think this context is adequately described in para8/p4 in the introduction/ background section of the paper.

9. Methods (page 8, last paragraph): Given the focus of policy changes on DM practices, the hypothesis of reduced injury rates seems unfounded. Please explain or clarify.

We appreciate the reviewer's careful reading of the manuscript. The outcome objectives of the strengthened disability management policy (para3/p8) were established by the organization (rather than the research team). Although this manuscript does not emphasize program activities focused on the prevention of work-related injury and illness, it was our view that the research team should report

the organizational objectives with fidelity.

Reviewer: 2

Reviewer Name: Marc White

Institution and Country: University of British Columbia, Canada Competing Interests: None declared

The paper is very well written. It avoids specialized language and is easy to understand across different audiences. The conceptual framework and research design demonstrates the benefits of using mixed methods. The paper lacks information about process evaluation and the potential influence of the research design and feedback process on the steering committee and their commitment /engagement in the process. (Having attended a conference with a presentation of this work I noted the "participatory action research" component that may be a contributing factor to the evaluation success and important part of the implementation process. The feedback process to the steering committee (and leadership) is missing from methods, discussion, outcomes/limitations and this limits the ability for replication or the appreciation of the role of the evaluating process as a potent contributor to the implementation process.

We appreciate the reviewer's comments, particularly the observation that the participation of the research team as a member of the labour/management steering committee for a three year period may have contributed to a portion of the achievement of this organizational change process. Reviewer #1 raised similar interests in Comment #2. We hope the description provided on p12/para1 is a sufficient declaration of the involvement of the research team.

VERSION 2 – REVIEW

REVIEWER	William Shaw Liberty Mutual Research Institute for Safety, USA I am one of 33 adjunct scientists who serve in an advisory role at the Institute for Work & Health (the first author's institution). However, I receive no direct compensation or research funding from the Institute, and I am not working directly with any of the authors. Thus, I feel I can provide sound and objective professional judgment about the submitted manuscript without any competing interest.
REVIEW RETURNED	21-Apr-2017

GENERAL COMMENTS	Thank you for the very detailed response to reviewer comments and for making a number of changes to the manuscript. All of my concerns and questions have been adequately addressed or incorporated. No further suggestions. The authors should be congratulated for making an important contribution in the area of employer disability management policies.
---

REVIEWER	Marc White Department of Family Practice, University of British Columbia, Canada
REVIEW RETURNED	21-Apr-2017

GENERAL COMMENTS	The beginning of the sentence "An early systematic review of research evidence..." is awkward and could be improved. In 2005 or ??
---

	Revisions have improved understanding of methods and process. This is a great study - I look forward to future replication.
--	--